# Deep Variational Instance Segmentation

**Jialin Yuan**
CoRIS Institute
Oregon State University
yuanjial@oregonstate.edu

**Chao Chen**
Stony Brook University
chao.chen.1@stonybrook.edu

**Li Fuxin**
CoRIS Institute
Oregon State University
lif@oregonstate.edu

## Abstract

Instance segmentation, which seeks to obtain both class and instance labels for each pixel in the input image, is a challenging task in computer vision. State-of-the-art algorithms often employ a search-based strategy, which first divides the output image with a regular grid and generate proposals at each grid cell, then the proposals are classified and boundaries refined. In this paper, we propose a novel algorithm that directly utilizes a fully convolutional network (FCN) to predict instance labels. Specifically, we propose a variational relaxation of instance segmentation as minimizing an optimization functional for a piecewise-constant segmentation problem, which can be used to train an FCN end-to-end. It extends the classical Mumford-Shah variational segmentation algorithm to be able to handle the permutation-invariant ground truth in instance segmentation. Experiments on PASCAL VOC 2012 and the MSCOCO 2017 dataset show that the proposed approach efficiently tackles the instance segmentation task. The source code and trained models are released at https://github.com/jia2lin3yuan1/2020-instanceSeg.

## 1 Introduction

Recent years have witnessed rapid development in semantic segmentation [30; 33; 10; 20], i.e., classifying pixels into different object categories such as *car* or *person*. However, in order to fully understand a scene, we need to identify individual object instances, along with their semantic labels. This task, called semantic instance segmentation [13; 16; 26], is much more challenging, because (1) different instances may have similar appearances if they belong to the same category; (2) the number of instances is often unknown during prediction; and (3) labels of the instances are *permutation-invariant*, i.e., randomly permuting instance labels in the training ground truth should not change the learning outcome (Fig. 1).

For such permutation-invariant instance labels, one cannot directly train the model using conventional objectives such as the cross-entropy (CE) loss. One popular strategy is to combine detection and segmentation into a two-stage approach. One network generates object proposals, while another one classifies and refines each proposal [17; 24; 39; 11; 18; 28; 9; 42; 1]. To ensure all instances are segmented, these methods often need to generate a significant amount of proposals ($1,000 - 3,000$ per image), and many are based on a sliding window approach that is similar to a complete search on a low-resolution image with anchor boxes. These proposals are verified with an object classifier and a smaller but still significant amount ($200 - 2,000$) are sent to the second stage for classification and refinement. To improve the efficiency, some recent works remove the anchor boxes by directly dividing the output image into a regular grid cell and segmenting the object that is centered in each

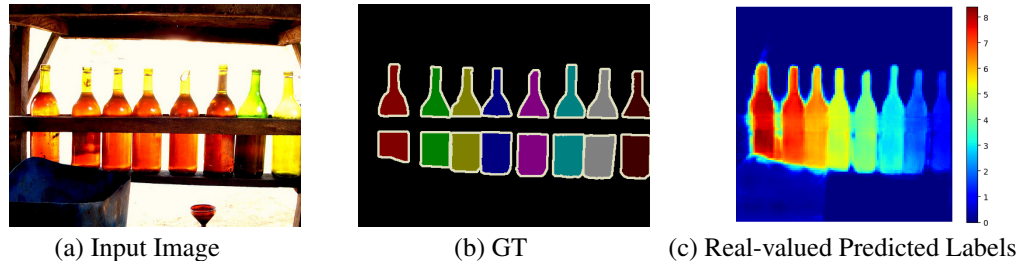

| (a) Input Image | (b) GT | (c) Real-valued Predicted Labels |

Figure 1: (a): An example from PASCAL VOC [13] with 8 bottles. (b) Ground truth. Labels of the bottles can be either 1 to 8 or 8 to 1. (c) Our approach solves a variational relaxation of the problem and predict real-valued labels on the image (best in color)

cell [45; 46; 47; 8]. However, they still require a significant amount of proposals. Another alternative solution is the search-free approach, which do not explicitly generate object proposals. Most methods learn to predict surrogates for instance labels for each pixel, and then use heuristic post-processing procedures to segment each instance [50; 49; 41; 3; 21; 27].

We note that the goal of instance segmentation is to generate piecewise-constant predictions on each pixel that match with a given ground truth. This resonates with the classic and elegant variational principle introduced to computer vision almost three decades ago. Such variational methods, originated from the Mumford-Shah model [32], parse an image into meaningful sub-regions by finding a piecewise smooth approximation. These approaches were traditionally limited to simple problems such as image restoration and active contours, mainly because the difficulties at that time to estimate nonlinear functions from an image. However, they could be inherently appealing in a deep network setting, since these variational objectives work with real-valued inputs and outputs. e.g., the Mumford-Shah functional, that are naturally differentiable.

We believe such variational approaches could be very powerful when combined with deep learning, since they enable us to solve deep learning problems that are difficult for conventional objective functions such as cross-entropy. On the other hand, parametrizing variational approaches with a deep network enables them to model complex functions originating from an image. It also allows them to generalize to testing images. In this paper, we propose *deep variational instance segmentation* (DVIS), which is a fully convolutional neural network (FCN) that directly predicts instance labels – a piecewise-constant function, with each constant sub-region corresponding to a different instance. A novel variational objective is proposed to accommodate the permutation-invariant nature of the ground truth in instance segmentation, which leads to end-to-end training of the network.

With this proposed approach, we are directly gazing at instances from a top-down FCN viewpoint without the need to generate bounding box proposals using search protocols. Our approach outperforms the other search-free instance segmentation methods on the PASCAL VOC dataset [13; 16] and it is the first search-free method tested on the MS-COCO dataset [26], obtaining a performance close to these search-based methods, but with significantly faster speed.

## 2   Related Work

Instance segmentation identifies every single instance at pixel-level. We groups the approaches tackling the task as search-based and search-free methods. Most search-based approaches are anchor based, they break the task into two cascaded sub-tasks: the first one generates region proposals with careful designed anchors boxes, e.g., with a region proposal network (RPN) [38]. Another network classifies and refines each proposal. This architecture solves the counting problem by adopting non-maximum suppression (NMS) [38; 36; 12; 29; 18; 19] or determinant point processes (DPP) [23; 2] to remove overlapping detections. Besides RPN, [43] uses selective search to generate proposals, [35] uses a network to generate region proposals in the form of a binary mask. However, such a search-base process is inherently slow, as many different proposals with various sizes and aspect ratios need to be generated and scored, which might be unacceptable in realistic application scenarios where engineers are striving to obtain real-time performance. [28; 9; 42] integrate instance-related features into the second stage in the anchor based architecture. The global context information encoded in these features can help refine the final segmentation. Recently, [6; 5] propose to use a

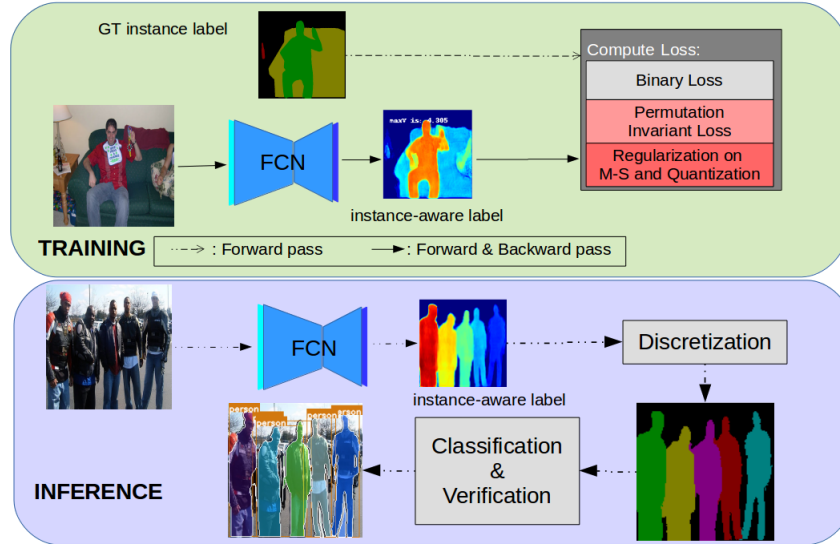

Figure 2: The proposed deep variational instance segmentation (DVIS): An FCN is trained to directly output real-valued instance labels, using a novel variational framework we proposed that combines a binary loss function, a permutation-invariant loss function, and regularization terms. During inference, we discretize the predicted instance map into several instances. After classification and verification, we output final segmentation with both semantic and instance labels (best viewed in color)

network to learn mask prototypes from the input image and combine these prototypes to generate the final mask for each detected instance. But they still search with anchor boxes of different scales and shapes hence generate significantly more proposals than ours. To reduce the redundancy of the anchor boxes, [45; 46; 47; 8] directly predict instance mask centered on each pixel in the output image. Instances that might share same centers are predicted at different scales from the FPN network [25].

We focus our literature review more on search-free methods that are directly relevant to our work. Some search-free approaches focus on exploring instance-aware and learning them using an FCN. [3; 39; 37] predict the energy of the watershed transform, [41] predicts the direction on each pixel to the object center, [21] predicts instance-level boundary score, and [27] attempts to locate instance segment breakpoints to separate each instance. However, these approaches do not directly generate an instance prediction and hence need to resort to a significant amount of heuristic post-processing such as template matching [41], MultiCut[21] or recurrent neural network[39; 37].

[22; 14] are search-free approaches based on the metric learning idea. [22] learns to map pixels to a multi-dimensional embedding space using pairwise associative loss. [14] formulates it using metric learning. The network is trained to enforce pixels from the same instance to be close to each other while pixels from different instance to be far away in the learned feature space. These approaches have not employed binary terms as in ours. Hence, in the embedding space generated by these methods, the background (stuff categories such as water, grass etc.) is no different than "yet another instance" and the separation between foreground and background is usually weak, hence these methods require more post-processing and depends on semantic segmentation to distinguish background and foreground, our foreground/background binary term directly suppresses output on the background pixels and outputs a cleaner instance map.

## 3 Deep Variational Instance Segmentation

### 3.1 The Mumford-Shah Model

The Mumford-Shah model is an energy-based model introduced in 1989 [32] for image segmentation. It relaxes the task to a continuous energy minimization problem that computes the optimal piecewise-smooth approximation of a given image. Let $I$ denote an observed image on a bounded domain $\Omega \subset \mathcal{R}^2$ to be segmented. We define $\hat{I}$ an approximation of $I$ and $C \subset \Omega$, the set of edges delineating

the boundaries of different objects. the Mumford-Shah functional is:

$$F(\hat{I}, C) = \int_\Omega (\hat{I}(x,y) - I(x,y))^2 dxdy + \mu \int_{\Omega \backslash C} |\nabla \hat{I}|^2 dxdy + \nu|C|, \qquad (1)$$

where $\mu, \nu$ are non-negative parameters, $\Omega \backslash C$ is the set of non-edge pixels, $|C|$ is the number of pixels in $C$. Minimizing the above functional essentially seeks to optimize for a piecewise smooth function (ideally constant inside each segment) which may be non-smooth on the edges/boundaries. The first term drives $\hat{I}$ to be close to $I$. The second term imposes smoothness prior inside each segment $\Omega \backslash C$ and protects from under-segmentation. The last term encourages shorter object contours to avoid over-segmentation. By adjusting the parameters $\mu, \nu$, it can optimally segment the given image.

The Mumford-Shah functional was well-regarded as a solid variational model that has been analyzed aplenty [7; 15; 34; 44; 48; 40]. It appropriately regularizes on the length of object boundaries while capable of modeling multiple objects within the same image. However, because the first term is usually only enforcing the approximation to be close to the input image function, it was traditionally only utilized in superpixel segmentation and active contours [44; 31].

**From unsupervised to supervised setting.** We note the similarity between the unsupervised Mumford-Shah model and the supervised instance segmentation problem. Both optimize for a piecewise-constant function, where each piece corresponds to one object instance and the number of pieces in the image is unknown. Both enforce constancy within each piece and a short boundary length would also be an ideal prior for instance segmentation, albeit to our knowledge we have never previously seen an approach that incorporates that. The second term in the MS-model is a common pairwise term that enforces piecewise-constancy, similar to those used in metric-learning-based instance segmentation methods [14; 22]. Previous work [48; 40] have shown that the second and third terms can be combined as a robust loss on the pairwise term (see Sec. 3.3 for more details).

The main difficulty of extending this variational approach to solve the instance segmentation problem lies in utilizing the matching potential $\int (\hat{I}(x,y) - I(x,y))^2 dxdy$, where a simple MSE or CE loss would not suffice for instance segmentation because of the permutation-invariance of ground truth labels. However, there is one ground truth label remains the same through the whole dataset: the background label. Thus, a new variational formulation is needed. In the next subsection we propose a novel variational formulation that solves the instance segmentation problem.

## 3.2 Deep Variational Instance Segmentation

As discussed above, we relax the supervised instance segmentation to a continuous energy minimization problem. We first note that the ground truth label $GT$ in instance segmentation usually has two distinct aspects: 1) when the label of a pixel is 0, then the pixel is background; 2) when the label of a pixel is larger than 0, then the label is *permutation-invariant*, i.e. one can switch labels of different objects (e.g. between object 3 and 5) without affecting their actual meaning. Hence, when defining a variational functional for instance segmentation, both of these components need to be considered.

We define a variational functional for instance segmentation as:

$$F(f, C) = \underbrace{\int_\Omega \mathcal{L}_b\left(f(x,y), \mathbb{I}_{[GT(x,y)=0]}\right) dxdy}_{\text{Binary Loss}} + \underbrace{\mu \int_\Omega \|\nabla f\|^2 dxdy + \nu|C|}_{\text{Regularization}} + \underbrace{\int_\Omega |f - Round(f)| dxdy}_{\text{Quantization}}$$

$$+ \underbrace{\int_\Omega \int_\Omega \mathcal{L}_{pi}\left(|f(x_1, y_1) - f(x_2, y_2)|, \mathbb{I}_{[GT(x_1,y_1) \neq GT(x_2,y_2)]}\right) dx_1 dy_1 dx_2 dy_2}_{\text{Permutation Invariant Loss}} \qquad (2)$$

where $f$ denotes the continuous-valued label map predicted by our network, an FCN with parameters $\omega$. $Round(\cdot)$ is the operation rounding to the nearest integer. $\mathcal{L}_b$ compares the instance label with the binarized ground truth label that indicates object/background and $\mathcal{L}_{pi}$ denotes the *permutation-invariant* loss function which compares the difference between two pixel labels $|f(x_1, y_1) - f(x_2, y_2)|$ with $\mathbb{I}_{[GT(x_1,y_1) \neq GT(x_2,y_2)]}$, which indicates whether the ground truth labels at these pixels are different. Using $\mathcal{L}_{pi}$, the exact values of the ground truth labels no longer play a role in the loss. The smoothness and minimal edge length terms are the same as in Mumford-Shah. We incorporate an additional quantization term, which drives the output label value to be closer to integers.

Training on this variational functional enables us to learn $f$ from a training set with instance-level ground truth and generalize onto unseen testing images. This improves over traditional variational segmentation which does not have learning capabilities. Note that in our permutation-invariant loss $\mathcal{L}_{pi}$, we would in principle integrate over *all* pixel pairs within the image that are not boundaries, instead of only in a small neighborhood as in traditional conditional random field (CRF) approaches. This is because instance segmentation is an inherently non-local problem: due to occlusion the same instance can be separated into several pieces in 2D that are possibly very far away from each other, hence, only local consistency is not enough. Empirically we have also found that if we only enforce local consistency, we may have small, smooth changes in the predicted instance labels $f$ that could add up to a significant amount and lead to changing instance labels within the same instance.

In practice we discretize $\mathcal{L}_b$ on all the pixels, and discretize the integral $\mathcal{L}_{pi}$ on sampled pixel pairs. Either stratified sampling or random sampling of pixel pairs can be used. In stratified sampling, we sample all the immediate neighbors in the 4-neighborhood of a pixel, and reduce the sampling density for further away pixel pairs. In random sampling, we randomly select pixel pairs across the whole image for computing the integral on $\mathcal{L}_{pi}$. We have found that on smaller resolutions, stratified sampling is efficient whereas when resolutions are very large, random sampling is more efficient.

Also note that there is a significant difference between variational approaches such as ours and CRF approaches, although both employ matching (unary) and regularization (pairwise) terms. In CRFs, the labels come from a discrete set, while in variational approaches the labels are relaxed to be continuous themselves. It is difficult for a CNN to simulate the full CRF inference process and one would have to resort to a recurrent network [51], increasing the complexity of the model. On the other hand, our variational formulation eq.(2) would only require an FCN to simultaneously handle images with an undetermined amount of objects, since it predicts labels as continuous real-valued numbers.

### 3.3 Loss Functions

As a variational approach, our output $f$ values are continuous. Hence, loss functions would be more similar to regression loss functions. Here we mostly utilize variants of the robust Huber loss function $L_h(v, \theta) = \frac{v^2}{2\theta}$ if $v < \theta$ and $v - \frac{\theta}{2}$ otherwise. We set $\theta = 0.1$ throughout the work.

**Binary Loss:** Our first $\mathcal{L}_b$ seeks to separate a labeled instance from "stuff" classes such as road, water, sky etc. which would not have individual instances in them and are usually labeled as background in instance segmentation tasks. Thus, $\mathcal{L}_b$ drives segmentation to be non-positive in background pixels and sufficiently positive in foreground pixels. Let $GT(x, y) = 0$ on the background pixels and $GT(x, y) > 0$ on the foreground pixels, the binary loss is computed as:

$$\mathcal{L}_b(f(x,y), GT(x,y)) = \begin{cases} L_h(ReLU(f(x,y))) & \text{if } GT(x,y) = 0 \\ L_h(ReLU(m_1 - f(x,y))) & \text{if } GT(x,y) > 0 \end{cases} \tag{3}$$

where $ReLU(x) = \max(x, 0)$ is the commonly used ReLU activation function, $m_1$ is a parameter of the loss function to separate foreground from background. With this loss, on foreground pixels, when $f(x, y) \geq m_1$, the loss will be 0, this accommodates foreground objects taking different $f(x, y)$ values. On background pixels, once $f(x, y) \leq 0$, the loss will be 0. In experiments, we set $m_1 = 2$. We formulate the term as regression with the robust Huber loss, instead of as binary classification with the CE loss. This is because the regression loss can obtain exactly 0 when the label value $\geq m_1$ on foreground and $\leq 0$ on background, whereas the CE loss tends to push to positive/negative infinity.

**Permutation Invariant Loss:** We use $\mathcal{L}_{pi}$ to enforce similarity between ground truth instance labels and predicted instance labels, taking into account that the ground truth labels are permutation-invariant. Let $p_1$ and $p_2$ be two pixels from a neighborhood and their ground truth as $GT_{p_1}, GT_{p_2}$, respectively, the relative loss is computed by:

$$f_d = |ReLU(f(x_1, y_1)) - ReLU(f(x_2, y_2))| \tag{4}$$

$$\mathcal{L}_{pi}\left(f_d, GT(x_1, y_1), GT(x_2, y_2)\right) = \begin{cases} L_h(f_d), & \text{if } GT(x_1, y_1) = GT(x_2, y_2) \\ L_h(m_2 - f_d), & \text{if } GT(x_1, y_1) \neq GT(x_2, y_2) \end{cases} \tag{5}$$

where $m_2$ is a parameter used to adjust the margin between predicted labels from different instances. We set $m_2 = 1$ in practice. Hence, there is no loss if the difference between predicted labels on two pixels is more than 1, which indicates that the two pixels belong to different instances. On the other

hand, if the two pixels belong to the same instance, the loss is 0 only when their predicted labels are the same.

**Regularization**: Mumford-Shah regularization is helpful for obtaining sharper boundaries. We have noticed that without such regularization the predicted label map tends to change more smoothly at object boundaries, creating intermediate values that do not belong to any object which make post-processing more difficult. There have been a significant amount of work on optimizing the Mumford-Shah term. We follow [40] to discretize Mumford-Shah as a robust loss function:

$$L_{MS}(f(x,y)) = \min(\mu\|\nabla f(x,y)\|^2, \nu) \tag{6}$$

which is equivalent to the original Mumford-Shah formulation. [40] then solves the formulation using a primal-dual algorithm, but in our case we do not need to exactly solve the optimization problem since optimization is anyways never exact with a deep network. Hence we just use a simple quasi-convex robust loss function as in the Cauchy loss:

$$L'_{MS}(f(x,y)) = \log\left((f(x,y) - f(x,y+1))^2 + (f(x,y) - f(x+1,y))^2 + 1\right) \tag{7}$$

Note one way to approach proper Mumford-Shah regularization is to anneal the loss gradually towards a Welsch loss function as in [4], which we did not do because the difference is very minor.

Finally, the quantization term minimizes the distance between the output label and its nearest integer. Gradient of this term is back-propagated from the first $f$. Since the operation $round(\cdot)$ is piecewise-constant, its gradient is 0). This term helps to create sufficient margin between different label values, making post-processing easier.

In summary, we relax a supervised instance segmentation to a deep variational minimization problem. With our formulation, the proposed variational problem can be tackled by training an FCN to optimize these loss functions and output the real-valued approximation of instance segmentation labels. And through directly optimizing on instance segmentation, our proposed approach has the advantage to generate different labels to different objects while has the capability of capturing multiple scattered parts, e.g. of an occlude sofa as a single object (Fig.2).

## 4 Implementation Details

**FCN for Instance Segmentation:** An encoder-decoder FCN network is adapted to solve instance segmentation with our variational loss. We employ *ResNet-50* and *ResNet-101* with output stride 8 as our base network and its output is then upsampled by 2 using a decoder network similar to the upsampling branch in FPN[25] to generate higher resolution output. The last layer of the FCN network outputs the real-valued label map as one output channel, which is then used to compute our variational loss eq. (2) and backpropagation. We remove negative label outputs by adding a *ReLU activation* on the FCN output. Note we did not employ multiple output heads as in FPN.

**Training:** We scale the input image to 513×513 for PASCAL and with the minimal edge equal to 700 for COCO (preserving the height-to-width ratio). The window size for computing relative loss is set to 128 throughout all experiments, except the ablation study about the parameter itself in supplementary. And we initialize the backbone network with the pre-trained weights for the semantic segmentation task on PASCAL and the pre-trained weights for the object detection task on COCO.

**Permutation-Invariant Loss:** Given an input image in size $H \times W$ and the FCN with a down-sampling factor $d$, the output size would be $\frac{H}{d} \times \frac{W}{d} \times 1$. The number of pixel pairs is a huge number $\frac{HW}{d^2} \times \frac{HW}{d^2}$. In our model, with the binary loss to separate background and foreground, it suffices to only consider the pixel pairs locating on instances, which reduces the number of pixel pairs that need to be computed. Then we utilize the stratified sampling to sample pairs to compute the permutation-invariant loss. Given a pixel $(x,y)$ and the window size $w$, we sampled all pixels inside the center area with distance $c(c < r)$ and we select the rest pixels with a dilation rate of 'r', similar to dilated convolutions [10]. The base setting we use is $w = 129, c = 8, r = 8$.

**Discretization to instance segmentation:** After we obtain the real-valued instance labels, we apply the mean-shift segmentation algorithm on it with different bandwidthes, 0.9 and 0.4 to discretize it to two different label maps. Because $m_2$ is fixed to 1, bandwidth of 0.9 works well to separate objects the network believe is different. And when the network does not learn to separate the instances well enough, bandwidth 0.4 helps to segment the objects. these two bandwidth proves to be enough to generate all instance segments, which are then verified in the next module.

**Classification and Verification:** We utilize a classification network to verify the segments. It first takes CNN features from the bounding box of each predicted instance from the FCN with ROIAlign [18], and concatenate it with the predicted binary mask for the instance. We then run a small convolutional network with 7 layers that will classify each predicted instance into the pre-defined semantic categories. Besides, we have an IoU head [19] that attempts to predict the Intersection-Over-Union between the predicted instance with the ground truth instance that best matches it, using a Huber regression loss. Finally, we reject false positive instances by thresholding on the weighted sum of predicted confidences on the semantic classification and the predicted IoU. Influence of the IoU head is studied in the supplementary material (Supl.Sec.4). Note that we are only verifying on average $5 - 15$ segments per image, which is significantly less than previous approaches (Table 6), hence the overhead of this stage is very small (Table 5). Hence, this classification step does not impact our speed advantage out search-based methods.

## 5 Experiments

We evaluate the proposed approach for instance segmentation on the challenging PASCAL VOC dataset[13] on the *val* split and the SBD split[16], as well as the COCO dataset[26].

### 5.1 Datasets

**PASCAL VOC 2012** consists of 20 object classes and one background class. It has been the benchmark challenge for segmentation over the years. The original dataset contains 1,464, 1,449, and 1,456 images for training, validation, and testing. It is augmented by extra annotations from [16], resulting in 10,582 training images. The metric we use to evaluate on PASCAL is average precision (AP) with pixel intersection-over-union (IoU) thresholds at 0.5, 0.6, 0.7, 0.8 and 0.9 averaged across the 20 object classes. As there is no ground truth on the testing set, we use the *val* set to test.

**PASCAL SBD** is a different split on the PASCAL VOC dataset. In order to compare with [24; 6], we train a separate model on the training set of SBD and evaluate on its 5,732 validation images.

**COCO** is a very challenging dataset for instance segmentation and object detection. It has 115,000 images and 5,000 images for training and validation, respectively. 20,000 images are used as *test-dev* from the split of 2017. There are 80 instance classes for instance segmentation and object detection challenge. There are more objects in each image than PASCAL VOC. We train our model on the *train 2017* subset and run prediction on *val 2017* and *test-dev 2017* subsets respectively. We adopt the public *cocoapi* to report the performance metrics $AP$, $AP_{50}$, $AP_{75}$, $AP_S$, $AP_M$, and $AP_L$.

### 5.2 Comparison to the state-of-the-art

**Results on PASCAL VOC and SBD** are shown in Table 1 and Table 2 respectively. Our approach significantly outperforms search-free approaches SGN and Embedding [27; 22] on all mAP thresholds. The latter two are state-of-the-art metric learning approaches. Besides, on the SBD dataset we also outperformed well-regarded anchor-based approaches DIN and FCIS [1; 24] significantly (Table 2). The recent YOLACT [6] achieved slightly better results than ours on mAP at $50\%$ IoU, however our approach is significantly better than it at $70\%$ IoU, which require more precise segmentation of each object. We note that $50\%$ IoU is a quite low standard for segmentation since there can still be significant amount of segmentation errors at this threshold. Our better performance at a higher threshold shows that our variational approach is capable of segmenting objects more precisely, especially on objects of non-rectangular shapes. Some proposal free approaches such as DWT takes each connected component as an instance, hence they do not work well for many PASCAL VOC objects which are separated into several parts with occlusions. We significantly outperformed SGN which is known to be superior than DWT. Qualitative results are shown in the supplementary material (Supl. Fig. 5).

**Results on COCO** are shown in Table 3 and Table 4. One can see that with a search-free algorithm, we obtain performances very close to the two-stage mask R-CNN, trailing mainly on small objects, where a complete search over all pixels would understandably help. We outperform the state-of-the-art anchor-based approach YOLACT on AP with multiple settings on both the *val-2017* and *test-dev 2017* datasets. YOLACT-700 results are only available on *test-dev* hence we compare with YOLACT-550 on *val*. The authors have a more recent improvement, YOLACT++ where they used deformable

Table 1: $AP^r$ result on the PASCAL VOC 2012 *val.* set.

| Method | backbone | architecture | $mAP^r$ | | | | | $AP^r_{avg}$ |
|---|---|---|---|---|---|---|---|---|
| | | | 0.5 | 0.6 | 0.7 | 0.8 | 0.9 | |
| DIN[1] | PSPNet(Resnet-101) | anchor-based | 61.7 | 55.5 | 48.6 | 39.5 | 25.1 | 46.1 |
| SGN[27] | PSPNet(Resnet-101) | | 61.4 | 55.9 | 49.9 | 42.1 | 26.9 | 47.2 |
| DML[14] | DeepLab-v2(Resnet-101) | | 62.1 | 53.3 | 41.5 | - | - | - |
| Embedding[22] | DeepLab-v3(Resnet-101) | search-free | 64.5 | - | - | - | - | - |
| DVIS | Resnet-50-FCN | | 68.4 | 63.3 | 58.1 | 49.1 | 33.7 | 54.5 |
| DVIS | DeepLab-v3(Xception 65) | | **70.3** | **68.0** | **60.2** | **50.6** | **33.7** | **56.6** |

Table 2: $AP^r$ result on the PASCAL SBD *val.* set.

| Method | backbone | architecture | $mAP^r$ | | | | | $AP^r_{avg}$ |
|---|---|---|---|---|---|---|---|---|
| | | | 0.5 | 0.6 | 0.7 | 0.8 | 0.9 | |
| DIN [1] | PSPNet(Resnet-101) | anchor-based | 62.0 | - | 44.8 | - | - | - |
| FCIS[24] | Resnet-101-C5 | | 65.7 | - | 52.1 | - | - | - |
| YOLACT[6] | Resnet-50-FPN | search-based | **72.3** | | 56.2 | | | |
| DVIS | Resnet-50-FCN | search-free | 70.0 | 67.0 | 61.0 | 49.1 | 27.8 | 55.0 |
| DVIS | DeepLab-v3(Xception 65) | | *70.5* | **68.5** | **62.9** | **55.2** | **34.5** | **58.3** |

convolutions which is orthogonal to our contributions, and could be applied in our case to further improve performance. Moreover, in Table 4, speed analysis on a V100 GPU (all post-processing included) is shown in the column *FPS*. Our method runs faster than all other baselines under the same backbone. The ResNet50 DVIS runs at $38.0\ fps$ and has $AP = 32.6\%$. Qualitative results are shown in the supplementary material (Supl. Fig. 6-7).

Table 3: $AP^r$ result on COCO's *val 2017* set

| Method | backbone | architecture | AP | $AP_{50}$ | $AP_{75}$ | $AP_S$ | $AP_M$ | $AP_L$ |
|---|---|---|---|---|---|---|---|---|
| PANet[28] | Resnet-101-FPN | | 37.6 | 59.1 | 40.6 | 20.3 | 41.3 | 53.8 |
| Mask R-CNN[8] | Resnet-101-FPN | anchor-based | 36.5 | 58.1 | 39.1 | 18.4 | 40.2 | 50.4 |
| YOLACT-550[6] | Resnet-50-FPN | | 30.0 | - | - | - | - | - |
| SOLO-800[45] | Resnet-50-FPN | | 36.0 | 57.5 | 38.0 | - | - | - |
| SOLO-800[45] | Resnet-101-FPN | search-based | - | - | - | - | - | - |
| PolarMask-800[47] | Resnet-101-FPN | | 29.1 | 49.5 | 29.7 | - | - | - |
| DVIS-700 | Resnet-50-FCN | search-free | 32.6 | 53.4 | 35.0 | 13.1 | 34.8 | 48.1 |
| DVIS-700 | Resnet-101-FCN | | **35.7** | **58.0** | **37.5** | **14.7** | **38.6** | **50.6** |

## 5.3 Ablation Study

**Inference cost**: We report the total number of float point operations (FLOPs) needed to compute instance segmentation with our approach compared with the state-of-the-art on the COCO *val2017* set.

In Table 5, it shows that our model requires significantly less computation than YOLACT[6], the state-of-the-art in inference speed, due to the fact that we have much less segments to work on (see also next paragraph and Table 6). We also present breakdowns of DVIS timings, where it can be seen that the majority of our computation is within the FCN network itself. Besides the network, the mean shift grouping and the classification module together only require about extra $2\%$ in terms of FLOPs.

**Number of Candidates in Post-Processing:** We compare the average number of candidates from our discretization process with previous one or two-stage instance segmentation algorithms in Table 6. All the search-based (even anchor-free) algorithms [24; 28; 47] send over 200 proposals to their second stage. SOLO [45] selects top-500 and YOLACT [6] selects top-200 proposals for post-processing. Meanwhile, we only average about $5-15$ segments per image sent to the classification module, further illustrating that our search-free FCN network has already precisely located the instances, thanks to the variational framework.

**More ablations in Supplementary material:** In the supplementary, we show that (1) the number of instances DVIS predicts is usually adequate in COCO (Supl. Sec. 1); (2) a large window size is important (Supl. Sec. 2); and (3) the MS loss generally only affects performance at the boundary but not the IoU. The quantization loss has benefits both on the boundary (Supl. Sec. 3) and on the IoU; (4) DVIS can predict instances that do not belong to any training categories (Supl. Sec. 5).

Table 4: $AP^r$ result on COCO's *test-dev 2017* set

| Method | backbone | type | AP | $AP_{50}$ | $AP_{75}$ | $AP_S$ | $AP_M$ | $AP_L$ | FPS |
|---|---|---|---|---|---|---|---|---|---|
| PANet[28] | Resnet-50-FPN | | 36.6 | 58.0 | 39.3 | 16.3 | 38.1 | 53.1 | - |
| FCIS[24] | Resnet-101-C5 | anchor- | 29.5 | 51.5 | 30.2 | 8.0 | 31.0 | 49.7 | 9.5 |
| Mask R-CNN[18] | Resnet-101-FPN | based | 35.7 | 58.0 | 37.8 | 15.5 | 38.1 | 52.4 | 13.5 |
| YOLACT-700[6] | Resnet-101-FPN | | 31.2 | 50.6 | 32.8 | **12.1** | 33.3 | 47.1 | 28.7 |
| SOLO-800[45] | Resnet-50-FPN | search- | 36.8 | 58.6 | 39.0 | 15.9 | 39.5 | 52.1 | 12.1 |
| SOLO-800[45] | Resnet-101-FPN | based | 37.8 | 59.5 | 40.4 | 16.4 | 40.6 | 54.2 | 10.4 |
| PolarMask-800[47] | Resnet-101-FPN | | 32.1 | 53.7 | 33.1 | 14.7 | 33.8 | 45.3 | 12.3 |
| DVIS-700 | Resnet-50-FCN | search- | 30.3 | 48.6 | 33.0 | 11.0 | 33.2 | 46.1 | 38.0 |
| DVIS-700 | Resnet-101-FCN | free | **32.9** | **52.6** | **34.6** | **12.5** | **36.7** | **48.1** | 30.4 |

Table 5: Number of FLOPs on the COCO *val 2017* set

| Method | backbone | FLOPs | |
|---|---|---|---|
| | | 550 | 700 |
| YOLACT[6] | Resnet-50-FPN | 61.59 G | 98.89 G |
| YOLACT[6] | Resnet-101-FPN | 86.05 G | 137.70 G |
| DVIS | Resnet-50-FCN | 38.49 G | 60.94 G |
| DVIS | Resnet-101-FCN | 66.24 G | 106.35 G |
| Breakdown for Postprocessing time on DVIS (ResNet-101) | | | |
| Mean Shift Grouping | - | 94.79 M | 124.42 M |
| Classification Module | Resnet-101-FCN | 1.54 G | 2.44 G |

Table 6: Number of candidates inputted to post-processing

| Method | No. |
|---|---|
| FCIS[24] | 2,000 |
| PANet[28] | 1,000 |
| Mask R-CNN[18] | 1,000 |
| YOLACT[6] | 200 |
| SOLO[45] | 500 |
| PolarMask[47] | 3000 |
| DVIS@ PASCAL VOC | 4.15 |
| DVIS@ COCO | 14.83 |

## 6 Discussion and Conclusion

In this paper we proposed deep variational instance segmentation (DVIS), which relaxes instance segmentation into a variational problem with a novel variational objective that includes a permutation-invariant component. Such a variational objective leads to an end-to-end training framework with an FCN directly predicting real-valued instance labels on the image. During inference time, we discretize the predicted continuous labels and utilize a small CNN to categorize them into semantic categories, as well as reject false positives. Experiments have shown that the proposed approach improves over state-of-the-arts in search-free instance segmentation approaches, especially on higher overlap thresholds, while being much faster. Such performance shows that our model is effective and efficient in capturing the global shape information in objects and segmenting object with higher precision.

DVIS showed a distinct philosophical difference from most search-based algorithms in that it is inherently processes the entire image with a single global glance. Most search-based algorithms look carefully at each local region to locate small objects, whereas DVIS directly gazes at the entire image and extract objects in one shot. Hence, DVIS might be missing out on some small objects, as our COCO results have shown. However, we argue that there are plenty of applications e.g. in robotics where segmenting the prominent objects quickly and accurately are of the utmost importance, rather than an exhaustive list of small and far-away objects. In those scenarios, the fast global approach of DVIS would make more sense since it deals with significantly smaller amount of object candidates. In the future, we will further explore variants of the top-down instance segmentation paradigm from DVIS to improve its performance on small objects.

## Broader Impact Statement

Instance segmentation is an important part for object recognition and is expected to be deployed in many real-life computer vision applications. Our algorithm significantly reduces the amount of computation required to obtain good performance in instance segmentation, hence would significantly lower the total carbon footprint for deployments of instance segmentation algorithms. We did not create additional social and ethical concerns of instance segmentation algorithms. However, there are inherent concerns about object recognition algorithms including instance segmentation to be misused in a system to recover personal identities without individual consent. This is beyond the scope of the paper since we are only concerned with broad object categories (person, trees, cars, bus, etc.) rather than individual identities of the objects. Our labels are permutation-invariant, i.e. they could assign an arbitrary real-valued number to any instance it predicts. Due to this randomness they do not reveal

individual identities per se. A possible concern is that one could input instance segmentation results to another algorithm to identify personal identities, however that is beyond the scope of this paper.

## Acknowledgement

The research was partially supported by NSF IOS-1546900, NSF IIS-1911232, IIS-1909038, an Amazon Research Award, DARPA Contract N66001-19-2-4035HR001120C0011 and a gift from Kuaishou Inc.. Any opinions, findings and conclusions or recommendations expressed in this material are those of the author(s) and do not necessarily reflect the views of the Defense Advanced Research Projects Agency (DARPA).

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
