[Supplementary Material]

# Supplementary materials of Deep Variational Instance Segmentation

## 1 How many labels can DVISpredict?

In the paper section 5.3, we give the average amount of candidates in post-processing and it is much smaller than RPN[7] based methods[1, 4, 3, 5]. Then an interesting question raised which is how many distinct objects can our framework predict. With multiple objects in the scene, the network has to be able to "see" all the objects, in order to assign them different values. Fig. 1 shows the number of candidate segments inputted to post-processing on the PASCAL VOC and MS-COCO dataset, which showed that our number of candidates are usually slightly higher than the number of objects. This showed that DVIS could both detect enough objects for each image, and also did not generate an overabundance of candidate segments.

Figure 1: Number of Objects DVIS predicted vs. number of objects in the image on Pascal VOC(the left column) and COCO (the right column). The figures are (from top to bottom): histogram of the number of ground truth objects in the dataset and the number of discretized instances over the number of GT objects. Note that by using 2 set of thresholds we are capable of detecting more objects than the maximal prediction value. And the number of candidate segments is only slightly more than the number of objects in the images

## 2 Window size for computing relative loss

We show an ablation study to verify that it is indeed necessary in the permutation-invariant loss to compare pixel labels with a large spatial displacement. The ablation study is done on the PASCAL VOC dataset. We compared results where we limit the permutation-invariant loss to pixel pairs that are close-by, with ranges of 8, 16, 32, 64, and 128 pixels tested respectively. Table 1 shows that a large window size significantly improves our performance.

Table 1: $AP^r$ result on PASCAL VOC val. set for different window size taken for the permutation-invariant loss

| Method | $mAP^r$ | | | | | $AP^r_{avg}$ |
|---|---|---|---|---|---|---|
| | 0.5 | 0.6 | 0.7 | 0.8 | 0.9 | |
| range 8 | 63.98 | 57.74 | 50.54 | 36.48 | 14.23 | 44.59 |
| range 16 | 63.38 | 57.55 | 49.72 | 37.49 | 14.09 | 44.45 |
| range 32 | 65.4 | 59.7 | 51.4 | 39.8 | 15.7 | 46.4 |
| range 64 | 68.21 | 62.82 | 56.73 | 49.34 | 33.5 | 54.1 |
| range 128 | **70.3** | **68.0** | **60.2** | **50.6** | **33.7** | **56.6** |

## 3 Regularization and Quantization

Since Mumford-Shah regularization term and the quantization term mostly work on improving the boundaries, their impact on the interior of the object is relatively small. Unfortunately, the commonly used IoU metric is almost exclusively focused on the interior and ignores small differences on the boundaries. Hence to illustrate the use of the MS-regularization, we compute the F1-measure, a semantic contour-based score from [2], to depict the effect of the Mumford-Shah regularization.

$$P_i^c = \frac{1}{C} \sum_{c=1\sim C} \frac{1}{M} \sum_{k=1\sim M} [d(z_{i,k}, GT_i^c) < \theta]$$

$$R_i^c = \frac{1}{C} \sum_{c=1\sim C} \frac{1}{M} \sum_{k=1\sim M} [d(z_{i,k}, GT_i^c) \geq \theta]$$

$$F_1 = \frac{1}{N} \sum_{i=1\sim N} \frac{2 \cdot P_i^c \cdot R_i^c}{R_i^c + P_i^c}$$

Where $i, c, m$ indicates the $m$-$th$ object in image $i$ with class $c$. $\theta$ is the distance error tolerance. The $[\cdot]$ is the Iversons bracket notation. $M$ is the number of objects with class $c$ in image $i$. $C$ is the total number of supported categories. $N$ is the number of images. From Table 2, the model trained with $\mathcal{L}_{MS}$ is 2% better than the model w/o $\mathcal{L}_{MS}$ at 1 distance error tolerance, which shows it improves significantly performance near the boundary. The model trained with adding quantization has equivalent performance with the model without it and it has higher score with larger distance error tolerance, since this term can increase margin between different instances and the detected instances are better shaped. Fig.2 shows some visual examples, the predicted instance map is more smooth, both inside the instances and on the background. Besides, instance boundaries are sharper with $\mathcal{L}_{MS}$. And different instances are better separated from each other by adding quantization.

Table 2: semantic contour F1-score on PASCAL VOC *val.*

| $\theta$ | 1 | 5 | 10 |
|---|---|---|---|
| w/o $\mathcal{L}_{MS}$ | 21.6 | 59.1 | 69.6 |
| w/ $\mathcal{L}_{MS}$ | 23.5 | 59.6 | 69.9 |
| w/ quantization and $\mathcal{L}_{MS}$ | 23.3 | 60.2 | 71.7 |

## 4 Influence of the IoU head

We run an ablation study to identify how the classification confidence $S_{cls}$ and the predicted IoU $S_{iou}$ affect the results. The weighted sum is computed as $\alpha * S_{iou} + (1 - \alpha) * S_{cls}$ with $\alpha = [0, 1]$. Fig.3 shows that it achieves better mAP at $70\% \sim 90\%$ IoU as $\alpha$ increases, which means the predicted IoU can detect more objects in higher quality.

| RGB image | without $\mathcal{L}_{MS}$ | with $\mathcal{L}_{MS}$ | with quantization and $\mathcal{L}_{MS}$ |

Figure 2: This figure shows the predicted instance map from model trained w/o or w/ the Mumford-Shah regularization, where the previous one is smoother inside the instances and the background and there is less noise along instances' boundaries

Figure 3: Ablation study on how the IoU score affect the instance segmentation on PASCAL VOC *val*.

## 5   Predict instance map on unseen categories

Because our DVIS method learn to segment instances directly from instance-level ground truth, it can recognize 'objectness' for unseen categories by relating them to seen ones. We test it with running the model trained on PASCAL VOC *train set* on images containing unseen categories from the DAVIS challenge [6]. Examples are shown in Fig.4, which shows DVIS can recognize 'objectness' and segment the instances.

RGB image                          GT                predicted instance map
Figure 4: Predicted instance map on unseen categories from DAVIS challenge [6].

## 6   Qualitative Results on PASCAL VOC

We show some more qualitative results on the PASCAL VOC dataset in Fig.5.

## 7  Qualitative Results on COCO

We show some more qualitative results on the MS-COCO dataset in Fig.6 and Fig. 7. We also show some failure cases in Fig.8. In those failure cases, our method fails to predict a good instance map when the scene become too crowded.

Note that part of the reason the algorithm is failing on those crowded scenes may be because of the way COCO is labeled. As can be seen in 8, among all the persons in the scene, only some are labeled as persons while some are not. We hypothesize this confuses our algorithm more than the anchor-based algorithms, since our permutation-invariant loss looks globally at all pixel pairs, whereas anchor box based methods only analyzes locally within each box. It would be interesting if we run the algorithm on a dataset where instances are more consistently labeled.

(a)    (b)    (c)    (d)

Figure 5: Examples from Pascal VOC 2012 *val* subset. From left to right: Image, Ground Truth, Predicted Instance Map, Final Instance Segmentation from DVIS(best viewed in color)

| RGB image | GT | Predicted Instance Map | final Seg. |

Figure 6: This figure shows qualitative results on COCO *val2017* set, part(1)

| RGB image | GT | Predicted Instance Map | final Seg. |

Figure 7: This figure shows qualitative results on COCO *val2017* set, part (2)

| RGB image | GT | Predicted Instance Map | final Seg. |

Figure 8: Examples of inaccurate predicted instance maps with crowded objects on the COCO *val2017* set