[Reviews · NeurIPS 2020]

Review 1

Summary and Contributions: The authors proposea. fully convolutional method for instance segmentation, that does not require region proposals or anchors like almost all other work in this field. The author's method is inspired by the class Mumford-Shah functional, which the authors have modified for this task. The proposed method trains the network to directly predict continuous scalar values corresponding to the instance-ids of objects in the image, using a permutation-invariant loss function based on Mumford-Shah. Unlike CRF-based approaches, a separate inference stage does not need to be performed, meaning that the network is simple to run inference on.

Strengths: The proposed formulation is novel and interesting, as it is a fully-convolutional approach that can readily handle a variable number of instances per image. I think this is a new direction for tackling instance segmentation, and so it could inspire interesting follow-ups by the community. The results on COCO, Pascal VOC and SBD are also good.

Weaknesses: The authors often call their method a "one step approach" and criticism other methods for using "heuristic postprocessing". I don't think the authors should be making these comments as there is a separate network to classify the masks. And I don't really buy the justification at the end of Page 6 that the proposed method needs to verify less masks than a typical two-stage method (ie Mask-RCNN) and is thus a "one-stage" method. Rather, I think the authors could be highlighting more the fact that their method does not require any anchors or region proposals, as I think this is a strong argument to be making. I also think the paper could benefit from an analysis of the number of instances that the network can predict. Ie, if the network was trained with a maximum of K instances in the training set, can it correctly predict more than K instances at test time? How does the network perform when presented with object classes that were not in the training set? Does it just treat these classes as background, or does it have some notion of "objectness"? As the authors use a separate network to classify the masks from the first stage, it is not obvious if all the main network has any notion of class labels at all. I also think the paper would benefit from more ablations on the sampling of pixels for computing the variational loss. The authors mention on Page 4 and 5 that sampling is important to not only have local consistency and to be able to deal with occlusions. However, there are no experiments to substantate this. Additionally, is it necessary to treat background separately, and have a separate "binary loss" for it. I expect that it could be handled as yet another class with the permutation invariant loss term. However, one difference is that background classes, in the current formulation, are encouraged to be mapped to any negative value, whilst foreground objects are effectively encouraged to map to an interval of size 1. Does this make a significant difference for handling the background class?

Correctness: Refer to "weaknesses" above for my comments on 1) Claiming the proposed method is a "one-step" method 2) Justification for the sampling strategy used by the authors in computing the loss. In Table 1, the AP^r_{avg} of DIN is more than the proposed method. However, the proposed method achieves a higher mAP at every IoU threshold. So clearly something is wrong here

Clarity: The paper is written well. There are just some minor typos Abstract. Line 14. "efficiently tackle" -> "efficiently tackles" Table 6. "Number of candidates inputted to post-processing" -> "Number of candidates input to post-processing"

Relation to Prior Work: This is ok on the whole. Some points to note Page 1, Line 15. Spatial transformer networks [20] are not an example of a development in semantic segmentation. Page 1, Line 28. [39] is not a proposal based method. Page 3. Line 79. I would not describe [1,37,39] as performing heuristic post-processing as they are all trained end-to-end and don't need any post-processing at the output. In fact, these papers all propose permutation-invariant losses too, so are perhaps the most relevant to this paper.

Reproducibility: Yes

Additional Feedback: ==== Update after rebuttal: Thank you for the rebuttal. I think the authors should refer to the supplementary material in the main paper, and also try to move some of the important experiments into the main paper. The other reviewers, particularly R4, raised some very relevant points. Nevertheless, I am maintaining my original rating, as I think the authors have proposed an interesting new way of tackling instance segmentation.


Review 2

Summary and Contributions: The paper presents an instance segmentation strategy that first generates a set of class-agnostic segment candidates based on an FCN, and then performs classification and verification on those candidates for final output. The main contribution is its candidate generation stage, which employs a label permutation-invariant loss function based on the Mumford-Shah functional to enforce piece-wise smooth outputs on the image plane. The proposed method is evaluated on the PASCAL 2012, SBD and MSCOCO dataset.

Strengths: - The idea of adopting variational energy minimization loss in generating instance segment candidates is interesting. It seems to generate a segment proposal set with higher quality and much smaller size, which makes the subsequent classification/verification more efficient. - The empirical evaluation shows promising performances compared to other one-stage IS methods. - The paper is mostly well written and easy to follow.

Weaknesses: - While the proposed method seems more efficient than many prior two-stage methods, it does not fit into one-stage category of IS. The overall instance segmentation pipeline still consists of two stages and in particular, the first stage does not predict the instance class label (and confidence score). The second stage of classification and verification requires a separate training step (Sec. 4). Hence the overall training is not end-to-end. - Some of its model design are not clearly explained: + The loss function in (2) introduces a rounding operator, which seems non-differentiable (or zero gradient a.e.) + For Binary loss in Sec 3.3, is there any particular reason to use a regression loss in Eqn (3)? Isn't it simply a binary classification task (CE loss)? + For Permutation invariant loss, Eqn (5) seems to miss a ReLU function? - The permutation invariant loss function has difficulty in handling large scale variation. For small objects, it has to densely sample the pixel pairs, which makes the computation very expensive. The variational regularization term also imposes a bias to favor objects with short boundaries, which can be restrictive. - The experimental evaluation is a bit lacking in several aspects: + As this method also uses two stage in instance segmentation, it should make comparisons with other SOTA two-stage methods instead of mostly one-stage methods. + For comparison on PASCAL benchmarks, it would be more fair to use the same backbone networks as the prior methods. In Table 1 & 2, the experiment results include many different settings, which makes it difficult to draw a conclusion. + It is unclear how well this method can generalize to crowded scenes with a large number of small objects, such as the street scene scenario. It would be more convincing if the paper can provided evaluation on other related benchmarks such as Cityscapes.

Correctness: See above.

Clarity: See above.

Relation to Prior Work: See above.

Reproducibility: No

Additional Feedback: ==== Update after rebuttal: The author rebuttal has addressed most of my original concerns with additional explanation on the model design, results with same backbone network and supplementary materials. Based on this, I would like to raise the rating to marginally above. On the other hand, I still think this method is closely related to proposal-based approaches and additional comparisons would make the paper's conclusion more convincing.


Review 3

Summary and Contributions: This paper tackles the problem of instance segmentation. In particular, the authors approach this task using a single stage network that outputs the instance segmentation masks as a bottom up approach. To do this, they use a loss inspired by the Mumford Shah variational method to handle multiple instances in a permutation-invariant way.

Strengths: - Single stage approach for instance segmentation using a variational method inspired loss, to my knowledge this is a novel approach that no one has tried before. It is great to see more work using/inspired by variational methods to tackle segmentation. - Good results with respect to other single stage models - Simple and novel permutation invariant loss to handle multiple instances - Model is much more efficient than most networks in the current literation in terms of the number of candidates that need post processing - Method/loss is simple and elegant

Weaknesses: - Would be good to compare/have a discussion with works using variational methods for segmentation. There seems to be a lot of missing related works in this field for example: End to end trainable active contours via differentiable rendering A deep level set method for image segmentation. Deep level sets for salient object detection Darnet: Deep active ray network for building segmentation Learning deep structured active contours end-to-end. Cnn-based semantic segmentation using level set loss Some of the papers I have listed here are explicit contour representations, it would be good to compare your model (implicit contour representation) with explicit ones. - It appears the model has difficulties with occlusion (Figure 1), could relaxing the hyperparameter on |C| in Eq 2 have improved this? - This paper lacks ablation studies to show the importance of the individual elements of the loss. For example, how important is the binary loss? How important is the quantization loss? How important is the regularization loss? This is something that I believe should be explored to have a better understanding of the contributions of the different losses. - The result is still significantly weaker than its two stage counterparts, on COCO it is 1.2 AP worse than mask rcnn and on cityscapes it is 3 AP worse - What is the run time speed of this network? Is it real time?

Correctness: Yes

Clarity: The paper is clear and well written.

Relation to Prior Work: No, see weaknesses section

Reproducibility: Yes

Additional Feedback: It is great to see more work using/inspired by variational methods FINAL REVIEW: I have read the other reviewers and the author's rebuttal and my original rating remains the same. I think the novel and simple approach to this problem is enough for this paper to be accepted. Though the performance is not the strongest I think this is a nice avenue for future interesting research.


Review 4

Summary and Contributions: This paper proposes a novel framework for instance segmentation. The framework is different from the current two-stage detection-segmentation pipeline, but adopts a segmentation-classification pipeline. Specifically, the authors claim there are mainly two problems. First is getting piecewise-constant results. Second, the piecewise-constant result should be permutation-invariant to the annotation. To solve those two, the authors propose a novel loss function. The authors solve the first problem by looking back to classical segmentation methods, especially the Mumford-Shah function, and transfer it into the instance-seg setting in a background segmentation sense. So, this part of loss can distinguish the fore- and back-ground pixels. For the second, this work uses the permutation invariant loss to segment the foreground area into instance pieces. To get the final bounding boxes and classification results, the authors use a classifier only in testing.

Strengths: 1. This paper proposes a novel pipeline for instance segmentation in a segmentation-classification sense, which is very valuable and inspiring. 2. This paper proposes a novel loss function for giving piecewise-constant predictions which are invariant to annotation permutations. This can be the basis for the following research. 3. The authors discuss the binary loss term and the permutation invariant term in detail. Taking advantage of the consistent background label is natural and effective. Comparing predictions in a pixel-wise way to deal with the permutation also matches human intuition. 4. The performance of the network on COCO and PASCAL is relatively good.

Weaknesses: 1. The authors claim this proposed framework is one-stage, but it is not correct. It is only “one-stage” in the training time. To get the bounding box and classification results, there must be a classifier. Thus, this pipeline just reverses the order of classification and segmentation from former “two-stage” methods. So, the framework in this paper should be defined as a “bottom-up” method, and mask-RCNN-like methods are “top-down” methods, where the “top” is object instances awareness, and “bottom” is pixel-wise understanding of the object instance like segmentation mask. 2. Reversing the order of two sub-tasks in the instance segmentation pipeline can make some significant changes, and it is a problem worthy of discussion. The first is the end-to-end trainable problem. If the classifier is only used in test-time, the whole network is not end-to-end trained, and getting bbox from mask is also not differentialble. The authors should give further discussion. 3. Another missed discuss is the performance of the proposed framework on objects with different scales. This should be the core difference between the top-down and bottom-up genres, but is ignored by the authors. The detection algorithm in top-down methods can sometimes perform badly on small objects (miss-detected). But the proposed bottom-up pipeline can have pixel-level segmentation results first, and may have better recall if the segmentation result is ideal. So, I think the author should add more material here. There should be quantitative or qualitative comparison between this work and top-down methods, with more fine-grained precision-recall curves on objects with different scales, rather than just a mAP for all object classes and scales. 4. Besides the whole pipeline, there are also some small problems in the loss function part. The first is the lack of ablation study. Like the regularization term, which is claimed to be helpful for better object boundary result, but without any quantitative and qualitative comparison, or citation. Same with the quantization term. 5. The second problem in the loss function part is that the authors miss some points in the discussion. If adopting a bottom-up pipeline, the ability of the segmentation network to capture intra-instance consistent and inter-instance different features is the key, and the loss function is actually designed for this. But the author seems to have ignored this perspective. An example is the regularization term. Making the prediction change smoothly is important for inter-instance consistency. And the abrupt change in instance boundary is the intrinsic clue for inter-class difference. The authors should give more discussion here. 6. The permutation-invariant loss seems require to maintain a large memory, e.g., NxN. It worth to describe how it is implemented in detail. 7. The comparison in Sec.5.2 is not fair. The author should use the same backbone with others. DeepLab-v3 is more powerful than PSPNet. After rebuttal: The authors addressed most of my questions, e.g., providing quantitative results using the consistent backbone, explaining the permutation-invariant loss. Considering the performance (compared to the current proposal-free methods, e.g., YOLO-based framework) and the novelty, I'd raise the rating to (5).

Correctness: The methodology part is mostly correct, except for the issues raised in the weaknesses.

Clarity: This paper is generally well written and understandable.

Relation to Prior Work: The related work regarding to different instance segmentation methods, e.g., top-down, bottom-up, direct, are not thoroughly discussed. There are also a lot of recent work missing, e.g., SOLO v1 and v2, PolarMask, PointRend, Centermask, etc.

Reproducibility: No

Additional Feedback:

[Author Response · NeurIPS 2020]

# Deep Variational Instance Segmentation

Thanks for the constructive feedback. We note most reviewers support the novelty of our new paradigm (**R1**, **R3**, **R4**) and the results (**R1**, **R2**, **R4**). We are encouraged that they found our work can inspire future directions (**R2**,

Table S1. $AP^r$ result on the PASCAL VOC and SBD *val.* set.

| Method | backbone | split | $mAP^r$ | | | | | $AP^r_{avg}$ |
|--------|----------|-------|------|------|------|------|------|------|
| | | | 0.5 | 0.6 | 0.7 | 0.8 | 0.9 | |
| DVIS | Resnet-50-FCN | VOC | 68.4 | 63.3 | 58.1 | 49.1 | 33.7 | 54.5 |
| DVIS | Resnet-50-FCN | SBD | 70.0 | 67.0 | 61.0 | 49.1 | 27.8 | 55.0 |

**R4**). **R3** recognizes our framework is simple and elegant. In the following we will address their specific comments.

**Whether it's one-stage (R1, R2, R4) or bottom-up (R4) or should compare against two-stage approaches (R2)**
We agree with **R1** that being anchor-free/proposal-free is a significant advantage of DVIS, as even all the latest "one-stage" approaches verify/predict at least 100-200 masks per image, whereas we only verify less than 20 (Table 6 + refs from R4). We'll revise the paper to emphasize the proposal-free aspect instead of one-stage (a minor revision as all "two-stage" prior work were anchor-based). Based on this, we argue that our performance shouldn't be judged against two-stage work which required significantly more computation than ours (**R2**). We respectfully disagree with **R4** that DVIS is "bottom-up" whereas anchor-based methods are "top-down". Note all CNN papers include a bottom-up backbone, hence even anchor-based methods are not simply top-down. We'd rather note that anchor-based methods are more local since they crop/predict on local regions, whereas we directly predict on the entire image by upsampling from a **low-resolution** feature map which possesses less local information on small objects. We envision DVIS to be used as a "quick global scan" which quickly finds the most prominent instances (more similar to human vision) whereas anchor-based approaches make an exhaustive local search. We will incorporate more discussions in the final paper.

**Ablations (Number of instances, effect of window size, unseen categories, individual loss functions (R1, R2, R3, R4)?** Those ablations were already presented in supplementary material Sec. 1-5. We showed that (1) the number of instances DVIS predicts is usually adequate in COCO (Supl. Sec. 1); (2) DVIS can predict instances that do not belong to any training categories (Supl. Sec. 5); (3) a large window size is important (Supl. Sec. 2); and (4) the MS loss generally only affects performance at the boundary but not the IoU. The quantization loss has benefits both on the boundary (Supl. Sec. 3) and on the IoU. Note that on DAVIS, DVIS can predict instances from similar categories as PASCAL (e.g. other animals), but have difficulty on ones that are vastly different from the training (e.g. ropes). We believe this is a matter of training data, and indeed there is an objectness factor that the network learned.

**The binary loss: Why use it (R1), why not a CE loss (R2), how important (R3).** Note that a binary loss is simpler and much faster to compute than the permutation-invariant loss (**R1**). The common cross-entropy (CE) loss tend to penalize differently based on the predicted real-valued labels whereas the robust Huber loss is exactly $0$ when the label value $\geq 2$ on foreground and $\leq 0$ on background. Hence it won't punish different predicted labels differently (**R2**). Binary loss is very important – without it we couldn't train the network properly hence there was no ablation (**R1**,**R3**).

**Speed analysis (R1, R3).** We don't possess a Titan-Xp hence can't report runtime consistent with literature. On our GTX 1080 Ti GPU, DVIS with ResNet-101-FCN is $20.5\%$ faster than YOLACT-700 with ResNet-101-FPN. According to the YOLACT paper, this should correspond to 28.6 FPS which would be real-time on a Titan-Xp.

**The permutation-invariant loss(R2, R4).** The memory requirement will indeed be large if all pixel pairs were considered. Hence we use an approach similar to dilated convolution to sample exponentially less neighbors that are further away (Sec. 4 ln. 222-229). In the end, for each pixel only 160 neighbors were considered, which is computationally tractable. We appreciate **R2** for reminding us that Eq. (5) misses a *Relu* function which will be fixed.

**Performance w.r.t. Instance size (R2, R4)** $AP_S$, $AP_M$, and $AP_L$ on COCO were reported in Table 3-4, which represent $AP$ for small, medium, and large objects respectively. Note our performance discrepancy from Mask R-CNN is the highest on small instances, different from R4's intuition (also see answer to the first question). Suppl. Fig. 8 shows some failure cases from COCO *val.*, which were mostly on crowded scenes with small objects.

**Different backbone in Table 1 and Table 2 (R2, R4).** We ran one more experiment on PASCAL using ResNet-50-FCN as the backbone. As shown in Table S1, results are slightly worse than the DeepLab-v3 backbone, but it still outperforms all competitors in Table 1 and outperforms YOLACT at 0.7 mAP (Table 2), consistent with our claims.

**References (R3, R4).** We'll cite the related work. Note that none of the work R3 mentioned were about the instance segmentation task. And the work R4 mentioned are very recent, published around after our submission at CVPR/ECCV 2020. None of these work are very similar to our approach. PointRend is proposal-based, SOLO/CenterMask are similar to YOLO in dividing the image into a regular grid and detect the object that is centering in each cell. PolarMask predicts surrogates, extending [3,39,37]. None of them predict instance labels directly from an FCN as our algorithm.

**The rounding operation in quantification loss (R2).** The rounding operation is piecewise-constant hence we set its gradient to $0$. Only gradient on the first $f$ from the quantization term is back-propagated to train the FCN.

**The $AP^r_{avg}$ of DIN in Table 1 (R1).** Upon inspection, we found that the average of the $AP^r$ in DIN is averaged on IOU thresholds ranging from 0.1 to 0.9, while others range from 0.5 to 0.9. We'll fix this number.

[Meta-Review · NeurIPS 2020]

The proposed formulation is novel and interesting. The empirical evaluation shows promising performances compared to other one-stage IS methods. However, the reviewers agree that it is closely related to proposal-based approaches and additional comparisons would make the paper's conclusion more convincing. Still, in the discussion the reviewers come to the conclusion that the paper provides a valuable contribution that is marginally above the acceptance threshold.